# Uncovering the Professional Landscape of Clinical Research Nursing: A Scoping Review with Data Mining Approach

**DOI:** 10.3390/nursrep15080266

**Published:** 2025-07-24

**Authors:** Mattia Bozzetti, Monica Guberti, Alessio Lo Cascio, Daniele Privitera, Catia Genna, Silvia Rodelli, Laura Turchini, Valeria Amatucci, Luciana Nicola Giordano, Vincenzina Mora, Daniele Napolitano, Rosario Caruso

**Affiliations:** 1Department of Biomedicine and Prevention, University of Rome Tor Vergata, 00133 Rome, Italy; 2Allied Health Professions Directorate, Orthopedic Research Institute, 40136 Bologna, Italy; monica.guberti@ior.it; 3Direction of Health Professions, La Maddalena Cancer Center, 90146 Palermo, Italy; locascio.alessio@lamaddalenanet.it; 4Department of Cardiovascular, Neural and Metabolic Sciences, Istituto Auxologico Italiano, IRCCS, San Luca Hospital, 20149 Milan, Italy; d.privitera@auxologico.it; 5Professional Development, Continuing Education and Research Unit, Bambino Gesù Children’s Hospital (IRCCS), 00165 Rome, Italy; catia.genna@opbg.net; 6CEMAD—Fondazione Policlinico Gemelli, IRCCS, 00168 Rome, Italy; silvia.rodelli@policlinicogemelli.it (S.R.); laura.turchini@policlinicogemelli.it (L.T.); valeria.amatucci@policlinicogemelli.it (V.A.); luciananicola.giordano@policlinicogemelli.it (L.N.G.); vincenzina.mora@policlinicogemelli.it (V.M.); daniele.napolitano@policlinicogemelli.it (D.N.); 7Department of Biomedical Sciences for Health, University of Milan, 20133 Milan, Italy; 8Health Professions Research and Development Unit, IRCCS Policlinico San Donato, 20097 San Donato Milanese, Italy

**Keywords:** clinical research nursing, clinical trials, competencies, barriers, professional identity

## Abstract

**Background/Objectives**: Clinical Research Nurses (CRNs) have emerged as pivotal actors in the conduct, coordination, and oversight of clinical trials globally. Over the past three decades, the role of the CRN has evolved in response to the increasing complexity of research protocols, ethical standards, and regulatory frameworks. Originating as task-oriented support figures, CRNs have progressively assumed broader responsibilities that include patient advocacy, protocol integrity, ethical vigilance, and interprofessional coordination. By mapping the global literature on CRNs, this review will examine how their role has been defined, implemented, and evaluated over the past three decades. **Methods**: A scoping review was conducted using JBI methodology and PRISMA-ScR guidelines. The search covered the peer-reviewed and gray literature from 1990 to 2024 across major databases. Data analysis combined traditional extraction with topic modeling, Multiple Correspondence Analysis, and k-means clustering to identify key themes. **Results**: From the 128 included studies, four major themes emerged: clinical trial management, role perception and team integration, professional competencies and development, and systemic barriers. Despite formal competency frameworks, CRNs face inconsistencies in role recognition, unstable contracts, and limited career pathways. Emotional strain and professional isolation are recurrent. Over time, their functions have evolved from task execution to broader responsibilities, including advocacy and ethical oversight. However, no studies reported patient-level outcomes, revealing a critical gap in the evidence base. **Conclusions**: CRNs play a vital but undervalued role in clinical research. Persistent structural challenges hinder their development and visibility. Enhancing institutional support and generating outcome-based evidence are necessary steps toward fully integrating CRNs into research infrastructures.

## 1. Introduction

Over the past decades, the global expansion and growing complexity of clinical research have created the need for specialized healthcare professionals who can bridge the gap between clinical care and research demands [1]. Among these, Clinical Research Nurses (CRNs) have emerged as key figures in ensuring clinical trials’ scientific, ethical, and operational integrity [2,3]. Defined as the specialized practice of nursing that balances the protection of research participants with the fidelity to study protocols, clinical research nursing represents a unique intersection between biomedical science and patient-centered care [2,3].

Historically, the daily management of trials was the domain of physicians [4]. However, the rise in translational research—aimed at accelerating discoveries from the lab to the clinic—has introduced new complexities and risks for participants [5]. CRNs have increasingly assumed responsibility for managing these challenges, applying their clinical, regulatory, and relational competencies to maintain both research quality and participant safety [6].

Although the CRN role has been described in the literature since the 1960s—particularly in early chemotherapy and oncology trials [7]—the 1990s marked a turning point. The role became more visible during this period and was progressively recognized as a legitimate nursing subspecialty, despite the ongoing ambiguity in role denomination and definition [8,9]. This lack of standardization continues to hinder their full integration into healthcare systems and limits the ability to measure CRNs’ contributions to clinical outcomes, trial performances, or participant experiences.

Moreover, while some national and international bodies have outlined core competencies and role domains for CRNs [2,3,10,11], the degree to which these frameworks have been adopted across healthcare institutions remains inconsistent.

The role of the CRN has often been historically conflated with that of the more general Research Nurse, despite clear distinctions in responsibilities, competencies, and scope of practice. While the term “Research Nurse” may broadly refer to any nurse involved in research-related activities, the CRN holds a more specialized position, embedded within clinical trials and regulated research environments [2,3]. This conceptual ambiguity has contributed to inconsistent recognition and role delineation in both the literature and clinical practice, underscoring the need for a more nuanced understanding of CRNs and their evolving professional identity [12].

As a result, CRNs often work in settings where their responsibilities are unclear, their impact is difficult to assess, and their role lacks formal recognition—frequently under short-term, project-based contracts that contribute to professional insecurity and increased work-related stress [12].

To address these gaps, this scoping review aims to map the literature on Clinical Research Nurses, tracing how the role has been defined, studied, and developed across contexts and identifying key themes, trends, and areas for future research.

## 2. Materials and Methods

### 2.1. Design

This scoping review was conducted strictly with the Joanna Briggs Institute (JBI) methodology [13] and adhered to a pre-established protocol (https://github.com/mattiabozzetti/protocols). To ensure methodological rigor and transparency, this review follows the Preferred Reporting Items for Systematic Reviews and Meta-Analyses extension for Scoping Reviews (PRISMA-ScR) guidelines [14].

### 2.2. Research Questions

The primary research question is: “What is the current state of literature on the role of CRN?” To explore this broad question, several secondary research questions have been developed to address specific aspects of CRNs’ competencies, roles, and barriers. Below are the secondary questions focusing on key aspects of CRNs’ practice, competencies, and challenges (Table 1).

### 2.3. Population, Context, and Concept

The search strategy for this scoping review was based on the Population, Concept, and Context (PCC) framework, ensuring a structured approach to identifying relevant studies. The population of interest comprised CNRs engaged in clinical trial protocols globally. CRNs operate within a specialized area of professional nursing that requires balancing participant care with strict adherence to research protocols. The concept under investigation centered on the competencies and scope of practice of CRNs, two intrinsically connected dimensions that collectively define their professional roles. Competencies encompass the comprehensive set of skills and knowledge that shape and inform the scope of practice, which in turn delineates the specific responsibilities and functions of CRNs within clinical trials, without rigid limitations. This study considered contexts involving CRNs working in clinical trial centers, metropolitan hospitals, and suburban healthcare settings worldwide, focusing on delivering research-related healthcare services to adult and pediatric populations. These clinical trial units were responsible for essential activities, including study design, participant recruitment, data management, dissemination, and analysis of randomized controlled trials and other rigorously structured investigations across various disease areas.

### 2.4. Eligibility Criteria

A wide range of study designs were considered, including qualitative, quantitative, mixed-methods, case studies, descriptive studies, peer-reviewed articles, theses, editorials, and position statements. No language restrictions were applied.

### 2.5. Search Strategy

A comprehensive search strategy was implemented to identify both published and unpublished studies between 1990 and 2024 (Appendix A). The process followed three steps. First, a preliminary search was conducted in MEDLINE (PubMed) and CINAHL (EBSCO) in July 2024 to identify relevant studies and extract key terms from titles, abstracts, and index terms. These informed a second, broader search across MEDLINE (PubMed), CINAHL (EBSCO), Web of Science (Clarivate), and PsycINFO (Ovid). To capture the gray literature, additional searches were performed in ProQuest Dissertations and Theses, OpenGrey, and on official CRNs’ organization websites, limited to active sites in English, Italian, and French as of 2024. The main search terms included combinations of keywords such as “clinical research nurse”; “research nurse”; “clinical trial nurse”; “competencies”; “barriers”; and “scope of practice”.

All references were managed with EndNote© V21, with duplicates removed.

### 2.6. Study Selection Process

Six reviewers conducted the screening independently, organized it into pairs, and distributed the records evenly among the teams. Each pair screened titles, abstracts, and full texts using Rayyan QCRI. In cases of disagreement within a pair, a third reviewer was consulted to resolve the conflict and reach a final decision. Although the process was rigorous, key challenges included managing the large volume of initial records and maintaining consistency across reviewers, which were addressed through regular team meetings and consensus-building discussions.

### 2.7. Data Extraction

For each study included in the review, data were systematically extracted across key categories, including author, year of publication, country, study design, objectives, population, concept, and context. In this initial phase, study results were recorded in an extended narrative format, providing a detailed summary for each article (Appendix A).

Data mining techniques—specifically Latent Dirichlet Allocation (LDA)—were employed to enhance the analysis beyond the traditional qualitative synthesis. The use of LDA was driven by the need for a more objective, data-driven approach capable of identifying underlying patterns and topics within a large and heterogeneous body of text [15]. Unlike manual coding or framework-based analysis, which can introduce researcher bias, LDA applies probabilistic modeling to detect hidden thematic structures in unstructured textual data, offering a systematic exploration of complex information.

### 2.8. Data Analysis

Following the initial data extraction phase—during which key information such as author, year of publication, country, study design, research focus, objectives, sample, and main findings were systematically collected—a lexicometric analysis of the textual corpus was performed to prepare the dataset for advanced exploration. The main results extracted from the studies were processed in an R 4.3.3 software [16] using LDA to identify latent thematic structures within the literature. Before running the LDA, we described the corpus’s lexicometric characteristics and applied the ldatuning package [17] to define the optimal number of topics, evaluating model performance through four established metrics [18,19,20,21]. These topics were confirmed by examining the highest-frequency terms and the visualizations produced through word clouds, which supported the interpretability and semantic coherence of the model outputs.

Recognizing that LDA, as a probabilistic model, can provide valuable insights but may not fully account for relationships between study characteristics, we applied Multiple Correspondence Analysis (MCA) and k-means clustering to validate and enrich the findings [22]. MCA was particularly useful for exploring the positioning of studies in a multidimensional space, allowing us to assess the alignment of the LDA-derived topics with key variables such as publication year, country, and study design. This additional layer of analysis contributed to evaluating the robustness and consistency of the thematic patterns identified.

The k-means clustering not only confirmed the thematic structure identified through LDA and validated via MCA, but it also enabled the identification of more granular subtopics (Figure 1). By grouping studies based on shared textual and metadata features, it provided a finer resolution of the thematic landscape, highlighting nuanced areas of convergence and divergence within the broader domains of CRN research. Full details of the data mining procedures and validation steps are available in Appendix A.

## 3. Results

As depicted in Figure 2, the PRISMA 2020 flow diagram guided the study selection process. Quantitative studies accounted for 47% (*n* = 54), followed by editorials (38%, *n* = 48), qualitative studies (13%, *n* = 22), mixed-methods (2%, *n* = 1), theses (2%, *n* = 1), and position statements (2%, *n* = 2). The most prevalent term is Clinical Research Nurse, which appears 73 times, indicating a strong dominance in the literature. This is followed by Research Nurse with 15 occurrences and Clinical Trial Nurse with 13, reflecting a narrower but still notable usage across sources.

### 3.1. Scope of Practice of CRNs (RQ1)

The scope of practice of CRNs emerged as one of the core thematic areas identified through topic modeling. The first latent topic extracted, “Role Perception and Team Integration,” accounted for 21% of the total corpus and captures how CRNs define their role within clinical research settings, particularly emphasizing interprofessional collaboration, integration into care teams, and advocacy for research participants [8,23,24,25,26,27,28,29,30,31,32,33,34,35,36,37,38,39,40,41,42,43,44,45,46,47,48]. This topic was characterized by high-frequency terms such as “integration,” “collaboration,” and “participant advocacy,” as revealed by the LDA outputs and word cloud visualizations (Appendix A). This evolution was further confirmed by MCA and clustering, which grouped these studies along a dimension associated with professional recognition and ethical responsibilities.

Closely related to this thematic area is the fourth topic, “Clinical Trial Management”, which accounted for 31% of the corpus and concerns the operational and procedural responsibilities of CRNs, including protocol adherence, data collection and monitoring, safety oversight, regulatory compliance [49,50,51,52,53,54,55,56,57,58,59,60,61,62,63,64,65,66,67,68,69,70,71,72,73,74,75,76,77,78,79,80,81,82,83,84,85], and ensuring the proper execution of clinical trials.

#### Role Perception, Integration, and Professional Recognition

A key subtopic identified through cluster analysis is the negotiation of role perception, integration, and professional recognition. CRNs are described as operating at the intersection of patient care and clinical research, balancing the demands of high-quality clinical practice with rigorous adherence to protocol and research standards [24,41,49,61,86,87,88,89]. Their professional scope has been formally articulated in some studies through five core dimensions: (i) clinical practice, (ii) study management, (iii) care coordination and continuity, (iv) human subjects protection, and (v) contribution to science [10,11].

Despite these frameworks, many studies report that the CRN role remains poorly defined and inconsistently recognized within institutional structures [28,31,35,36,90,91,92,93,94,95,96,97,98]. This lack of clarity is particularly critical in the advocacy work CRNs perform, such as safeguarding participant rights and ensuring ethical care during trial implementation [36,66,86,87,98,99].

Terminological ambiguity and overlapping role definitions often hinder CRNs’ integration into research teams, limiting their ability to function autonomously [25,100]. Insufficient professional recognition within research operations contributes not only to reduced visibility but also to emotional strain. Some sources associate the lack of formal acknowledgment with burnout [12] and moral distress [30], particularly when CRNs are held accountable for ethically sensitive decisions without the institutional support or clarity needed to navigate them effectively.

### 3.2. Core Competences of CRNs (RQ2)

The second topic, labeled “Professional Competencies and Development,” accounts for 25% of the corpus and reflects how the literature frames the knowledge, skills, and attitudes required for CRNs to perform their roles effectively [6,10,11,12,91,92,93,94,95,96,101,102,103,104,105,106,107,108,109,110,111,112,113,114,115,116,117,118,119,120,121].

#### 3.2.1. Formalization of Competencies and Reference Frameworks

The formalization of CRN competencies has been largely guided by two comprehensive frameworks: the Oncology Nursing Society (ONS) framework [3] and the International Association of Clinical Research Nurses (IACRN) framework [2]. Guided by these models, the literature analysis led to the identification of 965 competencies (Appendix A). Several studies extend and complement these frameworks by identifying additional or emerging competency areas, including facilities management, training delivery, project planning, regulatory knowledge, safety monitoring, and cross-disciplinary coordination [11,27,47,92,122]. The MCA and k-means clustering analyses placed these studies in close proximity to those addressing systemic challenges and career development, highlighting the conceptual overlap between competency acquisition and professional advancement. Notably, studies grouped in Cluster 3 (color-coded green in Figure 1) were more likely to report on institutional training gaps and the lack of protected time, pointing to structural barriers that hinder full competency deployment.

#### 3.2.2. Career Stagnation and Mismatch Between Skills and Professional Advancement

Despite the breadth and specialization of CRN competencies described in the literature, a notable gap exists between the expertise developed in practice and the opportunities for professional advancement. Many studies report that CRNs remain in the same position over extended periods, regardless of their accumulated experience and specialized skillsets [37]. This phenomenon is attributed to a lack of defined career pathways for nurses working in research settings and the perception, especially within hospital systems, that research activities are peripheral to mainstream clinical care. As a result, competencies that are highly valued in academic and trial settings—such as protocol navigation, regulatory compliance, or participant advocacy—are not necessarily recognized as meriting higher status, pay, or advancement opportunities in institutional hierarchies.

Several sources link this mismatch to broader structural factors, including unclear job descriptions, inconsistent training access, and the absence of protected time for role-specific development [6,93,102,106,108].

### 3.3. Obstacles Linked to the Position of CRNs (RQ3)

The obstacles associated with the role emerged as a prominent focus in the literature and were consolidated in the second topic extracted, “Barriers,” which accounted for 23% of the total corpus [86,87,88,89,90,97,98,99,100,122,123,124,125,126,127,128,129,130,131,132,133,134,135,136,137,138,139,140,141]. These themes were further articulated in three interrelated subtopics identified through cluster analysis: ethical tensions in clinical research practice, organizational and structural barriers, and systemic shortcomings in training and communication.

#### 3.3.1. Ethical Tensions in Care Delivery and Trial Implementation

The “Barriers” topic emerged as particularly dense and heterogeneous in the LDA analysis. Word clouds associated with this topic frequently highlighted terms such as “stress,” “contract,” “uncertainty,” and “support,” which were validated through the MCA’s positioning of the corresponding studies in Cluster 2 (blue in Figure 1). Cluster 2 reflects the ethical tensions that permeate the daily activities of CRNs, particularly in the context of recruitment, informed consent, and participant protection. CRNs are often placed in the position of defending participant autonomy while working under pressure to meet recruitment targets and ensure adherence to rigid trial protocols [27,59,60,91,133,141]. The literature describes this tension as ethically burdensome, especially in complex or high-stakes trials where the CRN is the primary advocate for the patient within the research team.

These tensions are frequently exacerbated by institutional hierarchies and sponsor-driven priorities that can limit the CRN’s ability to intervene on behalf of participants. In several studies, CRNs reported experiencing emotional fatigue and moral distress due to this structural imbalance of power and responsibility [87,97,98,124,126,127,130,131,132,133]. The ethical challenges described are systemic, recurring across research phases, and deeply embedded in the way trials are organized and executed.

#### 3.3.2. Organizational Barriers and Operational Constraints

In addition to identity and ethical challenges, organizational and operational limitations significantly affect the CRN practice. CRNs often work within rigid institutional frameworks, managing complex trials while facing limited training, coordination, and career development support. This highlights the combined impact of high workloads, insufficient educational resources, and extensive job responsibilities, which are further worsened by employment instability, such as temporary or project-based contracts, which restrict opportunities for long-term professional growth [33,94,96]. These structural challenges are reinforced by systemic weaknesses in training provisions and communication, showing how inadequate institutional support undermines the ability of CRNs to integrate effectively into multidisciplinary teams [6,12,23,91,102,105,106,108,109,110,122,123].

### 3.4. Current Employment Status of CRNs Worldwide (RQ4)

Employment conditions emerge as a critical and recurring concern throughout the literature, consistently situated within the broader theme of barriers. On one hand, studies emphasize structural challenges such as high workloads, temporary or project-based contracts, and limited institutional support. On the other hand, there is a recurring focus on the lack of clear career pathways, with many CRNs experiencing professional stagnation despite advanced skills and years of experience.

CRNs are frequently employed on short-term or project-based contracts, often tied to specific clinical trials or external funding mechanisms [33,94,96]. These arrangements, while financially pragmatic for institutions, create significant challenges for developing a stable and experienced workforce. Temporary contracts limit access to institutional benefits, continuing education, and long-term professional planning. In many cases, CRNs face uncertainty about contract renewal, role continuity, and workload redistribution at the end of each funding cycle, resulting in high job insecurity and stress levels [102,106,108,123].

These employment conditions also affect the recruitment and retention of professionals. The lack of stable career trajectories and limited institutional recognition make the role less attractive to new professionals and harder to sustain over time, especially in high-pressure or ethically complex research environments. Furthermore, the absence of dedicated career ladders for research nurses contributes to widespread stagnation, where CRNs remain in the same position for years despite accruing considerable expertise [37].

In many institutions, the CRN role is not formally differentiated from other nursing roles in terms of grading, pay scale, or advancement opportunities. This conflation reinforces the perception that research nursing is secondary to clinical care despite its increasing complexity and regulatory significance. Several studies also note the emotional toll of this professional invisibility, which may compound experiences of burnout already documented in relation to ethical and operational pressures [86,94].

### 3.5. Geographical and Temporal Patterns in the CRN Literature (RQ5 and RQ9)

The literature on CRNs has evolved over the past three decades, both geographically and thematically. Most studies were conducted in Europe (47%) and North America (40%), particularly the U.S. (38%) and the U.K. (31%), where the CRN is more institutionalized and recognized as a nursing subspecialty. In other regions, especially Asia and Oceania, the role remains less formalized but shows signs of growth.

Topic modeling and a biplot analysis (Figure 3) reveal that early publications, particularly from the U.S., focused on the procedural responsibilities linked to clinical trial management (Topic 4). From the mid-2000s, the literature shifted toward role perception and team integration (Topic 1), peaking around 2018. At the same time, studies from the U.K. and Northern Europe increasingly emphasized barriers (Topic 2), including institutional invisibility, ethical tensions, and employment precarity. Topic 3, related to competencies and training, remained stable across decades and regions, reflecting a sustained interest in formalizing the CRN profession. Together, these patterns show a clear transition from a task-based to a multidimensional understanding of CRNs, shaped by clinical, ethical, and systemic dynamics that vary across contexts.

### 3.6. Clinical and Organizational Settings in Which the Crn Role Has Been Implemented (RQ6)

Fifty-nine percent of the studies described the CRN role in general terms, without reference to a specific clinical setting. Of those that did, oncology was the most represented specialty (21%), likely due to the established presence of CRNs in cancer trials requiring complex, multi-phase protocols. In oncology, CRNs are involved throughout the research continuum, from enrollment to follow-up. Other specialties appeared less frequently, with renal care and pediatrics each at 3%, and pulmonology, pediatric oncology, and cardiology just above 2%.

### 3.7. Outcomes Used to Study the Impact of the CRN Role (RQ7)

Outcomes used to assess the impact of the CRN vary considerably across the literature and predominantly reflect operational, ethical, and professional dimensions. None of the included publications systematically examined patient-level health indicators (e.g., clinical improvement)—as direct outcomes of CRN activity.

The most frequently reported outcomes fall within the process and performance domains. Studies grouped under Topic 3 (Professional Competencies and Development) and Topic 4 (Clinical Trial Management) often assess variables such as adherence to protocol procedures, completeness of data collection, recruitment and retention rates, safety event reporting, and timeline compliance.

### 3.8. Temporal Patterns and Emerging Themes in the Literature (RQ9)

The temporal analysis of the literature shows a clear evolution in thematic focus across the four LDA topics (Figure 4). In the 1990s and early 2000s, Topic 4 (Clinical Trial Management) dominated, reflecting a view of CRNs as operational figures focused on compliance and documentation. From the mid-2000s, Topic 1 (Role Perception and Team Integration) gained prominence, marking a shift toward questions of identity, collaboration, and advocacy—peaking around 2018. Starting in the 2010s, Topic 2 (Barriers) emerged strongly, particularly in UK and European studies, highlighting structural and ethical challenges. In contrast, Topic 3 (Professional Competencies and Development) remained stable across decades, underscoring a persistent concern with training and role formalization.

## 4. Discussion

This review provides one of the most comprehensive and updated syntheses on the role of the CRN, combining broad geographical and temporal coverage with an innovative methodological approach. By integrating lexical data mining and automated topic modeling, it was possible to move beyond traditional narrative syntheses and identify latent patterns that have shaped the evolution of the CRN profession over the past three decades.

The analysis revealed four overarching thematic areas: operational trial management, role perception and team integration, advanced competencies and professional development, and systemic barriers. Together, these domains reflect a gradual yet significant shift in the literature—from a functionally defined, protocol-driven role to a more complex and autonomous professional identity encompassing clinical, managerial, and ethical dimensions.

This complexity is captured in the five domains internationally proposed by Bevans and Castro [10,11], which articulate the CRN’s role as operating at the intersection of patient care and research integrity. However, our findings underscore how the practical implementation of this model remains inconsistent. Many CRNs continue to work in contexts where their contribution is undervalued, their responsibilities are ambiguously defined, and their integration into clinical teams is partial or symbolic. These unresolved ambiguities generate emotional and cognitive strain, especially when CRNs must navigate ethically sensitive responsibilities—such as informed consent or participant protection—without adequate institutional support [142].

Indeed, the very absence of studies evaluating CRN-related clinical outcomes may be symptomatic of this conceptual vagueness. Where roles are ill-defined, it becomes difficult to isolate, evaluate, and legitimize their impact within scientific or managerial frameworks. Outcome measures in the literature are typically limited to procedural indicators (e.g., recruitment, data completeness, and adherence to protocol) which, although relevant, do not capture the broader relational or ethical contributions of CRNs to trial quality and participant wellbeing.

Structural and organizational barriers further complicate this scenario. CRNs are often employed under precarious contracts, with limited training and career progression access. This misalignment between the complexity of the role and the institutional resources allocated to support it undermines professional retention and the continuity and integrity of the research itself [143].

### 4.1. Research Gaps and Reccomendations

While this review reaffirms the centrality of the CRN’s advocacy role—particularly in relation to informed consent, emotional support, and ethical oversight—it also highlights a major gap: the near-total absence of research examining the clinical or patient-reported outcomes of the CRN practice. Despite frequent references to the importance of relational care, no studies to date have investigated how CRNs’ communicative and emotional competencies impact participant experiences or therapeutic processes in research settings.

Insights from the neurophysiological and psychosocial literature on the doctor–patient relationship suggest that relational dynamics—mediated through trust, empathy, and presence—can modulate therapeutic outcomes via identifiable biochemical pathways [144]. This opens a promising avenue of inquiry: whether and how the CRN–participant interaction activates similar mechanisms. Yet, this remains unexplored.

Beyond the general call for outcome-based evidence, each of the four thematic areas identified in this review suggests distinct and meaningful directions for future research. Regarding clinical trial management, there is a pressing need to investigate how a CRN’s presence, staffing levels, and workload distribution influence operational outcomes such as recruitment and retention rates, protocol adherence, data completeness, and overall trial timelines. These operational metrics are crucial not only to assess the contribution of CRNs but also to inform staffing models that enhance efficiency and reduce research waste.

With respect to role perception and team integration, future studies should explore how professional recognition—or lack thereof—affects team dynamics, job satisfaction, and role clarity. Particular attention should be given to how ambiguous role boundaries may contribute to emotional burden, ethical tension, and limited autonomy in decision-making processes. In the domain of professional competencies and development, further research is needed to understand how access to structured career pathways, continuing education, and protected time for research-related duties impact skill advancement, retention, and long-term engagement in the CRN role.

Lastly, the theme of systemic barriers highlights the importance of examining institutional determinants—such as leadership support, policy frameworks, and contractual stability—that shape the sustainability of the CRN practice. Very few studies to date have addressed how these structural variables influence both the effectiveness of the CRN and the overall quality of research delivery.

Moreover, the literature remains silent on whether and how CRNs’ relational and communicative competencies might impact clinical or patient-reported outcomes. This represents a promising avenue for interdisciplinary research, particularly in light of neurobiological evidence—such as Benedetti’s work—suggesting that empathy and trust can modulate physiological and psychological responses. Integrating both organizational and clinical perspectives will be essential to develop robust, evidence-based models for the optimal integration of CRNs in research settings.

### 4.2. Limitations

This review offers a comprehensive and methodologically innovative mapping of the CRN role; however, some limitations should be acknowledged. Despite the inclusion of a broad range of sources and formats, some relevant editorials or the gray literature may have been missed due to inconsistencies in indexing or database coverage.

## 5. Conclusions

This review identified four main thematic areas—trial management, role perception, competencies, and systemic barriers—revealing a progressive shift in the literature from procedural duties to a more complex and autonomous identity. Despite being described in the literature for decades, the role remains inconsistently defined, unevenly institutionalized, and rarely evaluated in terms of clinical impact. The analysis confirms that, while the CRN is an increasingly central figure in clinical research, the literature on this profession remains relatively underdeveloped, with major gaps in outcome assessment, role standardization, and cross-contextual comparisons. Continued conceptual and empirical work is needed to consolidate the role and capture its full contribution to both science and care.

## Figures and Tables

**Figure 1 nursrep-15-00266-f001:**
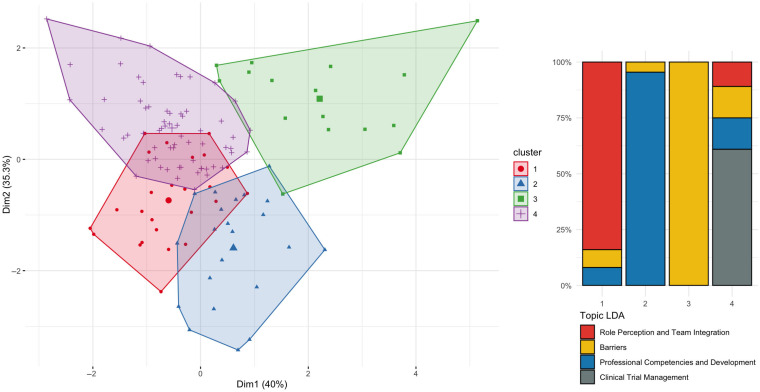
Distribution of included studies across four thematic clusters identified through MCA and k-means clustering (**left**), with corresponding topic proportions derived from LDA modeling (**right**). Each cluster is associated with a predominant thematic focus: (1) role perception and team integration (red), (2) barriers (blue), (3) professional competencies and development (green), and (4) clinical trial management (purple). The bar chart illustrates the dominant LDA-derived topics within each cluster, highlighting distinct thematic compositions.

**Figure 2 nursrep-15-00266-f002:**
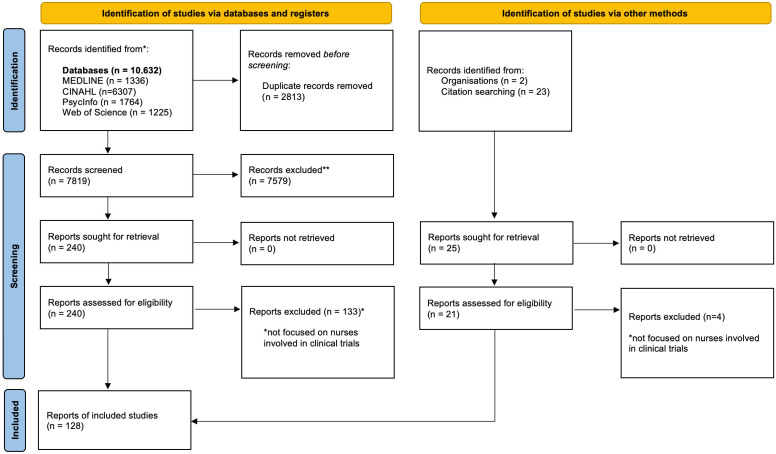
PRISMA 2020 flow diagram illustrating the identification, screening, and inclusion process of studies in the scoping review. * Not focused on nurses in trials. ** Records excluded during title and abstract screening primarily due to a focus on general nursing research rather than on nurses involved in clinical trials or due to insufficient methodological detail.

**Figure 3 nursrep-15-00266-f003:**
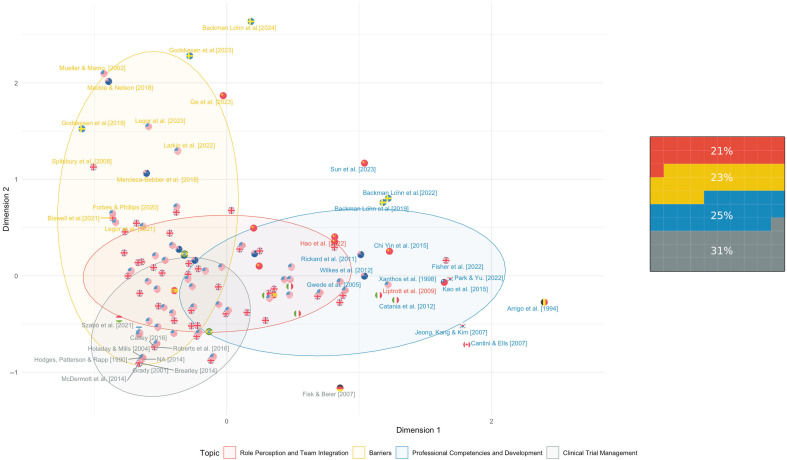
Panel (**left**): Biplot visualizing the distribution of included studies (*n* = 128) by dominant topic and year of publication. Each flag represents a study, color-coded by the main topic identified through Latent Dirichlet Allocation (LDA): role perception and team integration (red) [8,23,24,25,26,27,28,29,30,31,32,33,34,35,36,37,38,39,40,41,42,43,44,45,46,47,48], barriers (yellow) [86,87,88,89,90,97,98,99,100,122,123,124,125,126,127,128,129,130,131,132,133,134,135,136,137,138,139,140,141], professional competencies and development (blue) [6,10,11,12,91,92,93,94,95,96,101,102,103,104,105,106,107,108,109,110,111,112,113,114,115,116,117,118,119,120,121], and clinical trial management (gray) [49,50,51,52,53,54,55,56,57,58,59,60,61,62,63,64,65,66,67,68,69,70,71,72,73,74,75,76,77,78,79,80,81,82,83,84,85]. Labels highlight the most externally positioned studies. Ellipses indicate topic clusters with 95% confidence intervals. Panel (**right**): Proportional distribution of studies by dominant topic. The 10 × 10 grid represents 100% of the sample, with each square corresponding to 1%. Colors match those in the biplot.

**Figure 4 nursrep-15-00266-f004:**
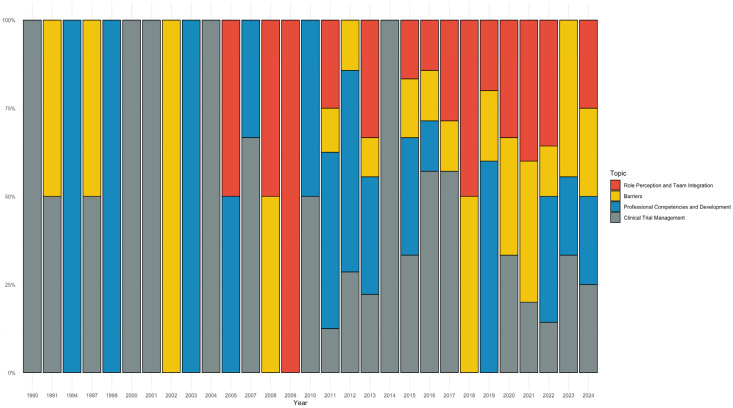
Temporal distribution of thematic focuses across the literature on CRNs, based on LDA topic modeling. The figure illustrates how dominant themes—such as clinical trial management, role perception and team integration, professional competencies, and systemic barriers—have evolved across different publication periods.

**Table 1 nursrep-15-00266-t001:** Secondary research questions.

Research Questions (RQ)	
RQ1	What is the scope of practice for CRNs?
RQ2	What are the core competences of CRNs?
RQ3	What obstacles are linked to the position of CRNs?
RQ4	What is the current employment status of CRNs worldwide?
RQ5	What geographical contexts have examined the role of CRNs?
RQ6	What clinical and organizational settings have contributed to developing and implementing the CRN role?
RQ7	What outcomes have been used to evaluate the impact of CRNs?
RQ8	What emerging themes or subtopics have been identified through data mining techniques?
RQ9	Is there a temporal pattern in the thematic focus of the literature, and are there emerging themes tied to specific publication periods?

Notes: CRN = Clinical Research Nurse.

## Data Availability

Full data are available upon reasonable request to the first author.

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
