# Peer review of "Uncovering the Professional Landscape of Clinical Research Nursing: A Scoping Review with Data Mining Approach"

_nursrep, 2025, doi:10.3390/nursrep15080266_

Round 1
Reviewer 1 Report
Comments and Suggestions for Authors
Dear authors, congratulations and recognize the hard work done in this study that denotes a scientific rigor according to its design. Undoubtedly, nursing in the field of research is essential, despite this it is well known among the professionals who are dedicated to it, that it is not adequately recognized either by the general population or by health institutions. Without a doubt, this study will contribute to making this field of nursing visible, contributing to its progress.
In general, the manuscript complies with the rules and guidelines associated with design, being the description of the objective of the study and its corresponding title, the aspect that has caused me the most disagreement. Nursing research, as is well known, covers more field of work than that of the defined study population: "nurses who participate in clinical trial protocols", consequently, I suggest redefining the main objective of the study, as well as the title assigned to it.
As minor comments I suggest: include the search terms in the main text; improve the appearance of Figure 1 by matching the colours of the clusters to the point areas; in the flow chart, for the first screening the explanatory note on the causes of exclusion is missing; Figure 3, whose content is already included in the text, could be deleted because it does not provide relevant visual information; Figure 4 should improve the visualization of the data; and Figure 5 requires a better description in the title, it is proposed to change it to a line graph.
Author Response
We thank the reviewer for the thoughtful and encouraging feedback on our manuscript. We are grateful for the recognition of the scientific rigor and potential impact of our work. We also appreciate the valuable suggestions regarding the title, objectives, and visual presentation of figures.
We have addressed all comments carefully:
-
Clarified the specific focus on Clinical Research Nurses (CRNs) in the introduction;
-
Added search terms in the main text (Section 2.5);
-
Revised Figures 1, 4, and 5 for clarity and coherence;
-
Added explanatory notes to the PRISMA flowchart (Figure 2);
-
Removed Figure 3 as suggested to streamline the manuscript.
We trust that these revisions improve the overall quality and readability of the paper. Detailed responses and changes are provided in the marked manuscript and response document.

Reviewer 2 Report
Comments and Suggestions for Authors
Consider adding more to your introduction and providing more context. Consider decreasing the reference list.

Author Response
Dear Editor,
We thank Reviewer 2 for the helpful and constructive comments. In response:
-
We have expanded the Introduction to provide greater conceptual and historical context around the role of Clinical Research Nurses (CRNs), clarifying their distinction from broader nursing research roles.
-
Regarding the reference count, we respectfully clarified that 128 references correspond to the included studies of the scoping review. Additional citations were necessary to contextualize methodological and conceptual aspects. We carefully reviewed all references to ensure that each is relevant and non-redundant.
We appreciate the reviewer’s feedback, which helped us refine the manuscript’s focus and rigor.
Kind regards,

Reviewer 3 Report
Comments and Suggestions for Authors
Dear Authors,
I read your manuscript with great interest. It presents a well-conducted scoping review of the scope of practice and competencies of Clinical Research Nurses over the past 30 years. The combination of innovative data analysis methods (LDA, MCA, k-means clustering) with established JBI methodology and PRISMA-ScR guidelines makes this work valuable both substantively and methodologically. The article is well-structured, written in a coherent and scientific language, and the conclusions are clearly supported by the data analysis.
I appreciate the comprehensive methodology, the broad scope of the reviewed literature, the identification of key themes, and the recognition of the lack of patient-level outcome reporting, which is crucial for future research directions.
Despite the high value of the manuscript, I have a few suggestions that could further strengthen the impact of this study:
-
In scoping reviews such as this one, it is standard practice to include a table with the extracted data from the analyzed articles. This table, often referred to as the “characteristics of included studies,” significantly enhances the transparency and credibility of the review.
-
In addition to the general call for outcome-based evidence, the discussion could be expanded to include more specific directions for future research stemming from each of the four identified themes (e.g., specific research questions concerning professional recognition, career development, or mitigation of emotional burden).
-
The included tables and modeling results are interesting but are not always discussed in sufficient detail.
-
The reader would benefit from an additional summary table presenting, for example, the number of publications by topic and country.
-
Figure 2 should be moved to section 2.5, as it essentially describes the search strategy.
This manuscript makes a significant contribution to the advancement of knowledge in clinical research nursing and is very well prepared. The suggested revisions are supplementary and do not require fundamental methodological changes.
Author Response
We thank Reviewer 3 for the thoughtful and encouraging feedback. We are pleased that the methodological rigor, clarity, and relevance of the manuscript were appreciated.
In response to the reviewer’s suggestions:
-
We provided a more detailed narrative discussion of the modeling results (LDA, MCA, k-means), particularly in the Results section, to enhance interpretability.
-
We expanded the discussion to offer specific future research directions stemming from each of the four identified themes, integrating both organizational and clinical perspectives.
-
To support transparency, the characteristics of included studies were placed in the Supplementary Materials, given their volume (n = 128).
-
Figure 3 was revised for improved clarity, and we retained Figure 2 in the Results section, consistent with PRISMA-ScR guidance.
We appreciate the reviewer’s valuable input, which helped us refine and strengthen the manuscript.
